# Towards Robust Concept Erasure in Diffusion Models: Unlearning Identity, Nudity and Artistic Styles

## Abstract

Diffusion models have achieved remarkable success in generative tasks across various domains. However, the increasing demand for content moderation and the removal of specific concepts from these models has introduced the challenge of *unlearning*. In this work, we present a suite of robust methodologies that significantly enhance the unlearning process by employing advanced loss functions within knowledge distillation frameworks. Specifically, we utilize the Cramer-Wold distance and Jensen-Shannon (JS) divergence to facilitate more efficient and versatile concept removal. Although current non-learning techniques are effective in certain scenarios, they are typically limited to specific categories such as identity, nudity, or artistic style. In contrast, our proposed methods demonstrate robust versatility, seamlessly adapting to and performing effectively across a wide range of concept erasure categories. Our approach outperforms existing techniques, achieving consistent results across different unlearning categories and showcasing its broad applicability. Through extensive experiments, we show that our method not only surpasses previous benchmarks but also addresses key limitations of current unlearning techniques, paving the way for more responsible use of text-to-image diffusion models.

## 1 Introduction

Diffusion models Ho et al. (2020); Dhariwal & Nichol (2021); Kawar et al. (2022); Gu et al. (2023) have advanced text-to-image generation, enabling the creation of high-quality visuals from diverse prompts. However, the extensive datasets used to train these models, often sourced indiscriminately from the Internet, pose significant ethical and safety challenges. It also presents a serious risk of misuse. Their ability to generate realistic images can be exploited to produce misleading or harmful content, such as deepfakes, disinformation, or unauthorized reproductions of copyrighted material. These concerns underscore the need for effective safeguards, including methods that can selectively *unlearn* or restrict certain concepts to prevent the misuse of these models.

Recently, there has been increasing interest in developing techniques to *unlearn* or *erase* specific concepts from diffusion models. In this direction, progress has been made in unlearning nudity, artistic, insignia, copyrighted images, and identity styles Kumari et al. (2023); Zhang et al. (2024a); Heng & Soh (2024); Gandikota et al. (2023); Kim et al. (2023); Golatkar et al. (2024); Lu et al. (2024). Although *concept erasure* is the primary objective of these methods, achieving consistent effectiveness in various unlearning scenarios remains a significant challenge. Identities of well-known figures, such as politicians or celebrities, are intricately woven into the latent space of the model, making them difficult to remove. Unlike simpler concepts such as nudity, which can often be managed by filtering outputs, identities are deeply embedded in the model's knowledge, requiring more sophisticated techniques for effective erasure.

**Motivation.** A widely used unlearning approach in diffusion models involves fine-tuning the original model by conditioning the noise estimate on the target concept to be removed, guiding it to align with the unconditional estimate Kumari et al. (2023); Gandikota et al. (2023); Kim et al. (2023). All the existing methods use the L2 loss function during fine-tuning. The choice of loss function is critical for guiding the model's ability to selectively remove undesirable concepts, such as nudity

or artistic styles, while preserving image quality in retain set data. We empirically observe that L2 loss is limited in handling complex, multi-modal distributions within the model's latent space. This motivated us to explore other loss functions such as Jensen-Shannon (JS) divergence and Cramer-Wold distance. Both provide a nuanced way to align the model's output distribution with the target distribution, allowing for more effective unlearning without catastrophic forgetting.

**Our Contributions.** In this work, we empirically demonstrate that employing L2 loss as the default loss function in the unlearning (fine-tuning) setup results in suboptimal performance in current unlearning methods. We study the impact of employing advanced loss functions such as JS divergence and Cramer-Wold distance within a knowledge distillation framework. We hypothise that L2 loss provides a simple pointwise correction, whereas, JS divergence and Cramer-Wold distance enable a more robust alignment of distributions within the latent space of diffusion models. These loss functions are better at handling complex, multi-modal latent spaces. This allows for the removal of targeted concepts while retaining the model's generative capabilities. By leveraging these loss functions, we ensure that unlearning is adaptable across different scenarios and minimizes interference with non-targeted concepts, resulting in a more versatile and reliable unlearning process.

We demonstrate how JS divergence and Cramer-Wold distance can be effectively integrated into the unlearning process to guide the removal of targeted concepts while preserving image quality in the remaining dataset. We provide comprehensive experiments showing that our approach not only outperforms existing L2-based methods in concept removal but also mitigates the risk of catastrophic forgetting, thereby maintaining the model's versatility across various unlearning scenarios. By introducing these loss functions in the knowledge distillation framework and demonstrating their effectiveness to erase concepts, we offer a significant step forward in the development of safe and reliable unlearning techniques for diffusion models.

## 2 RELATED WORK

We discuss some of the major challenges in ensuring effective moderation of content generated by text-to-image diffusion models. These challenges include handling NSFW and restricted content, addressing rights and usage concerns. We review how prior research has approached these issues.

**NSFW and Restricted Content.** Diffusion models can be misused to generate inappropriate images, including violent or explicit content. Some methods attempt to mitigate this by filtering training data or employing post-processing safety checks during inference Gandhi et al. (2020); Nichol et al. (2022); Bedapudi; Rando et al.; Schramowski et al. (2023). While effective, these methods can be bypassed when open-source code and model weights are publicly accessible Sharma et al. (2024). A more difficult-to-circumvent approaches involve altering the model's knowledge by modifying attention weights Gandikota et al. (2023), through fine-tuning Zhang et al. (2024b), or with continual learning Heng & Soh (2024). However, they have been shown to be prone to partial diffusion based attacks in Sharma et al. (2024). Our method proposes a more effective solution to erase unwanted concepts that also performs well in the recently introduced unlearning metrics Sharma et al. (2024).

**Rights and Usage Concerns.** Stable diffusion models trained on extensive datasets such as LAION-5B Schuhmann et al. (2022) have been found to infringe on copyrights related to artistic styles, largely because of their tendency to memorize copyrighted material in the training data Zhang et al. (2024c). Under U.S. copyright law [1], the concept of *Fair Use* allows others to use copyrighted material only if it is transformed in a way that differs significantly from the original. Legal precedents have shown that structural similarity to the original work can result in infringement claims.

Somepalli et al. (2023); Wang et al. (2024) examined how diffusion models can infringe on copyrights through memorization, enabling the generation of existing posters, artworks, and other images that may be protected by copyright. To address this, some studies have suggested adding minor perturbations, acting as watermarks, into images to prevent their memorization by diffusion models Cui et al. (2023); Zhao et al. (2023). However, these watermarks can be readily eliminated using techniques like denoising or blurring. Another approach involves post-training model adjustments to delete undesired concepts, such as modifying model weights to remove specific styles Gandikota et al. (2023); Heng & Soh (2024); Kim et al. (2023); Kumari et al. (2023); Zhang et al. (2024b).

---

[1]https://www.copyright.gov/title17/

**Unlearning in Diffusion Models.** Unlearning in diffusion models presents a challenge due to the intricate and interconnected nature of their latent space representations. Concepts within these models are not independent but are woven into a larger knowledge distribution. Consequently, the removal of specific elements, like copyrighted logos or identifiable human faces, without compromising the model's ability to produce high-quality images is difficult. Current leading methods in the field involve fine-tuning the diffusion model by adjusting attention heads or using distillation techniques that condition the noise estimate on the concept to be removed Gandikota et al. (2023); Heng & Soh (2024); Kim et al. (2023); Kumari et al. (2023); Zhang et al. (2024b). However, recent findings by Sharma et al. (2024) indicate that many of these approaches Kumari et al. (2023); Gandikota et al. (2023) do not completely erase the concepts, as they can still be generated using partial diffusion probing.

## 3 Preliminaries

### 3.1 Unlearning in Diffusion Models

Unlearning aims to remove specific learned concepts from diffusion models while preserving the model's overall performance. In diffusion models, the goal is to eliminate targeted knowledge, such as biases or harmful content, without degrading the image generation quality. The unlearning process can be expressed through the following general equation:

$$\theta_{\text{unlearned}} = \theta_{\text{fully trained}} - \eta \sum_{t=1}^{T} \nabla_\theta L_t(\theta; \mathcal{D}_r, x_t, c),$$ (1)

where $\theta_{\text{unlearned}}$ is the model's parameter after the unlearning process, $\theta_{\text{fully trained}}$ is the model's parameter before unlearning, $\eta$ is the learning rate, $T$ is the total number of diffusion steps, $c$ denotes the conditioning using a particular modality, and $L_t(\theta; \mathcal{D}_r, x_t, c)$ is the loss function at step $t$.

### 3.2 Unlearning via Distillation

Several existing methods have adopted different forms of knowledge distillation for unlearning in classification models, regression models, language models and diffusion models. We define the general form of the unlearning objective as:

$$\mathcal{L} = \mathcal{D}\left(\epsilon_{\theta_{\text{student}}}(z_t, c_s, t), \text{sg}(\epsilon_{\theta_{\text{student}}}(z_t, t))\right),$$ (2)

where $\mathcal{D}(\cdot, \cdot)$ is a generic distance function or loss metric. This could be the $L^2$ norm, $L^1$ norm, Kullback-Leibler (KL) divergence, or any other suitable loss function. Here, $\epsilon_{\theta_{\text{student}}}(z_t, c_s, t)$ is the noise estimate conditioned on the target concept $c_s$, and $\epsilon_{\theta_{\text{student}}}(z_t, t)$ is the unconditional noise estimate. The stop-gradient operation, $\text{sg}(\cdot)$, is applied to prevent gradient computation for the unconditional noise estimate, ensuring that updates are made solely based on the conditioned estimate.

The teacher model is then updated using the student's parameters to smooth out abrupt changes:

$$\theta_{\text{teacher}} \leftarrow \phi(\theta_{\text{teacher}}, \theta_{\text{student}}),$$ (3)

where $\phi(\cdot, \cdot)$ denotes the update rule that integrates the student's potentially drastic latent updates into a more stable adjustment for the teacher model. This update process reflects the student's learning while ensuring that the changes to the teacher model remain gradual and controlled, avoiding harsh updates that may destabilize the model.

## 4 Proposed Work

We formally define the key mathematical properties for three distances Cramér-Wold Cramér & Wold (1936) distance, Jensen-Shannon Divergence Lin (1991); Rao & Nayak (1985), and L2 distance experimented in this work. To our knowledge this is the first work to mathematically and empirically analyze the significance of each distances in the context of unlearning in diffusion models. We introduce four lemmas by analyzing several mathematical principles such as functional

analysis (norms and decomposition), measure theory (distributional differences), information theory (entropy and divergence measures), and orthogonal projections. We will use these principles to rigorously demonstrate the limitations of L2 distance and the effectiveness of JS divergence and Cramér-Wold distance in capturing concept-specific changes in a diffusion model.

### 4.1 LIMITATIONS OF L2 DISTANCE IN UNLEARNING

We identify two main limitations of using L2 distance in unlearning. First, we demonstrate that L2 distance is highly sensitive to irrelevant dimensions. To this end we formally show:

**Lemma 1.** *(L2 Distance is Sensitive to Irrelevant Dimensions)*
*Let $\mathbf{z}_T, \mathbf{z}_S \in \mathbb{R}^n$ be two $n$-dimensional vectors representing the teacher and student embeddings respectively, and let the concept $\mathbf{c}$ to be unlearned be represented in a subset of dimensions $\mathcal{D}_c \subseteq \{1, 2, \ldots, n\}$. The L2 distance between $\mathbf{z}_T$ and $\mathbf{z}_S$ is non-zero even when the concept is fully unlearned, as long as $\mathbf{z}_T$ and $\mathbf{z}_S$ differ in irrelevant dimensions, $\mathcal{D}_{nc} = \{1, 2, \ldots, n\} \setminus \mathcal{D}_c$.*

The proof for Lemma 1 shows that the L2 distance between vectors $mathbf{z}_T$ and $mathbf{z}_S$ can be decomposed into concept-related and non-concept-related parts. If the concept is fully unlearned, the contribution from concept-related components becomes zero. However, if differences exist in non-concept-related components, the overall L2 distance remains positive, indicating that the L2 distance is not zero even when the concept is unlearned. The detailed proof is added in the Appendix.

Next we show that L2 distance have difficulty capturing correlated conceptual changes, which is crucial property for any unlearned system.

**Lemma 2.** *(L2 Distance Fails to Capture Correlated Conceptual Changes)*
*Let $\mathbf{z}_T, \mathbf{z}_S \in \mathbb{R}^n$ be two $n$-dimensional vectors. Suppose the concept $\mathbf{c}$ is represented as a linear combination of multiple correlated dimensions in $\mathbf{z}_T$. If the unlearning process changes these dimensions uniformly in $\mathbf{z}_S$, then the L2 distance will overestimate or underestimate the actual conceptual difference.*

The Lemma 2 argues that if a concept $\mathbf{c}$ is represented by multiple correlated dimensions in $\mathbf{z}_T$, and these dimensions are uniformly altered during the unlearning process in $\mathbf{z}_S$, the L2 distance will not accurately reflect the true conceptual change. This is because L2 distance treats each dimension independently, failing to capture the correlation between dimensions. As a result, the L2 distance may either overestimate or underestimate the difference depending on the nature and scale of the uniform changes, the detailed proof can be found in appendix.

### 4.2 QUANTIFYING CONCEPTUAL DIFFERENCES WITH JS DIVERGENCE

We show JS divergence is well equipped to quantify conceptual differences.

**Lemma 3.** *(Jensen-Shannon Divergence Accurately Quantifies Conceptual Differences)*
*Let $\mathbf{z}_T$ and $\mathbf{z}_S$ be $n$-dimensional embeddings represented as probability distributions $P$ and $Q$ over the same $n$-dimensional space. Suppose the concept $\mathbf{c}$ is encoded in a subset of dimensions $\mathcal{D}_c \subseteq \{1, 2, \ldots, n\}$. If $P$ and $Q$ differ only in the concept dimensions $\mathcal{D}_c$ and are identical in the remaining dimensions $\mathcal{D}_{nc} = \{1, 2, \ldots, n\} \setminus \mathcal{D}_c$, the Jensen-Shannon (JS) divergence will accurately capture the concept difference while remaining invariant to changes in $\mathcal{D}_{nc}$.*

The proof for Lemma 3 demonstrates that the Jensen-Shannon (JS) divergence is a better measure for concept unlearning compared to L2 distance. It starts by defining the JS divergence between two probability distributions $P$ and $Q$, and decomposes these distributions into concept-related and non-concept-related components. By showing that if the non-concept-related parts of $P$ and $Q$ are equal, the JS divergence becomes zero for these dimensions, it highlights that JS divergence is invariant to changes in irrelevant dimensions. In contrast, L2 distance would still capture these differences, making it sensitive to irrelevant variations. Therefore, JS divergence accurately captures conceptual changes, while L2 distance may overestimate the differences. The detailed proof is provided in the Appendix.

Table 1: We compare the CLIP scores obtained by the CW, JS and L2 unlearning methods. We show the mean CLIP score by unlearning the following concepts: *baby, narendra modi, elon musk, amitabh bachchan, nike, nudity, pablo picasso, vincent van gogh). lower is better↓.*

| Steps | SD 1.4 | CW | JS | L2 |
|---|---|---|---|---|
| 1500 | 29.7618 | 23.4975 | **22.9702** | 23.4436 |
| 1400 | 29.7618 | 23.4934 | **22.6502** | 23.0212 |
| 1300 | 29.7618 | 24.1404 | **24.4098** | 23.4957 |
| 1200 | 29.7618 | 25.2973 | 24.8594 | **23.0008** |
| 1100 | 29.7618 | **25.5798** | 25.8554 | 26.3378 |

Table 2: Comparing the unlearning performance in terms of $\mathcal{CCS}$, and $\mathcal{CRS}$ score. We compare the CW and JS with L2 loss based method. We evaluate the effectiveness of concept erasure for three types of concepts: *celebrity, baby,* and *artistic style.*

| Concept Erased | Prompt | $\mathcal{CCS}$ | | ↑ | $\mathcal{CRS}$ | | ↑ |
|---|---|---|---|---|---|---|---|
| | | CW | JS | L2 | CW | JS | L2 |
| amitabh bachchan | amitabh bachchan | 0.74 | 0.61 | 0.74 | 0.05 | 0.05 | 0.04 |
| baby | baby with teddy bear | 0.62 | 0.62 | 0.63 | 0.02 | 0.02 | 0.03 |
| vincent van gogh | sunflowers by vincent van gogh | 0.51 | 0.42 | 0.52 | 0.009 | 0.008 | 0.008 |

### 4.3 CAPTURING HIGH-ORDER CONCEPTUAL CORRELATIONS WITH CRAMÉR-WOLD DISTANCE

Cramér-Wold distance has ability to capture higher-order correlations and joint distributional changes between dimensions, which can be formally represented as follows:

**Lemma 4.** *(Cramér-Wold Distance Captures High-Order Conceptual Correlations)*
*Let $\mathbf{z}_T, \mathbf{z}_S \in \mathbb{R}^n$ be two vectors, and let $P, Q$ be the corresponding distributions over these vectors. If the concept $\mathbf{c}$ is represented by correlations between multiple dimensions, then the Cramér-Wold distance between the distributions of $\mathbf{z}_T$ and $\mathbf{z}_S$ will be zero if and only if $\mathbf{c}$ has been fully unlearned, even if the individual L2 distances in some dimensions are non-zero.*

The proof for Lemma 4 establishes that Cramér-Wold (CW) distance is a more suitable measure for detecting conceptual changes compared to L2 distance. It begins by defining the CW distance as the supremum over 1-dimensional linear projections of the distributions $P$ and $Q$. It then shows that CW distance captures correlations by analyzing projections aligned with the concept subspace. If a concept is represented by correlated dimensions, changes in these dimensions will be detected by CW distance but may be missed by L2 distance, which only considers dimension-wise differences. Finally, the proof highlights that CW distance is invariant to changes in irrelevant projections, making it robust in measuring concept changes across different subspaces. The detailed proof is provided in the Appendix.

## 5 EXPERIMENTS AND RESULTS

We show that both Cramér-Wold distance and Jensen-Shannon (JS) divergence outperform L2 distance in a knowledge distillation setup for unlearning in diffusion models. Our approach is compared against existing state-of-the-art methods ESD-u, ESD-x Gandikota et al. (2023), Ablating Concepts Kumari et al. (2023), and SDD Kim et al. (2023), which also utilize a knowledge distillation framework but rely on L2 distance as their default loss function. We choose SDD Kim et al. (2023) to represent all four distillation-based unlearning methods, as it has already been demonstrated to outperform ESD-u, ESD-x, and Ablating concepts in Sharma et al. (2024). The L2 loss results are basically the results in SDD. We denote Cramér-Wold results as **CW**, JS Divergence results as **JS**, and L2 loss-based method as **L2** in our discussions.

Table 3: CLIP directional similarity score based comparison of the proposed CW and JS methods with L2 loss based method. Both CW and JS perform better erasure as compared to the L2 based unlearning. *higher is better↑*.

| Concept Erased | Prompt | CW | JS | L2 |
|---|---|---|---|---|
| baby | a baby playing with teddy bear | 0.1289 | **0.1874** | 0.1363 |
| narendra modi | narendra modi in a park | 0.1375 | **0.1876** | 0.0996 |
| elon musk | elon musk shaking hand with a girl | **0.2085** | 0.1802 | 0.1348 |
| amitabh bachchan | amitabh bachchan in blue background | 0.1682 | **0.2138** | 0.1738 |
| nike | a shoe with nike logo | 0.0926 | 0.0585 | **0.1058** |
| | mean score | 0.1328 | **0.1463** | 0.1284 |

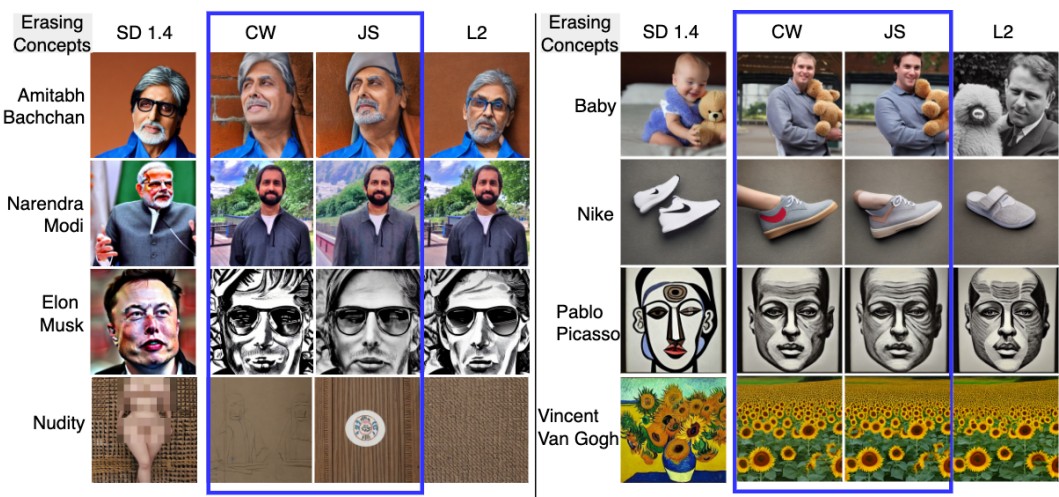

Figure 1: The concept erasure results are shown for the CW, JS, and L2 methods for a variety of erasure tasks. We observe similar results in most of the categories except in *Nike* concept erasure, where CW and JS generated images of shoe without *Nike* logo, whereas, L2 generated footwear that is not a shoe but looks like *crocs*.

**Evaluation of Concept Erasure.** We assess the effectiveness of unlearning using four metrics: the CLIP score, CLIP directional similarity score Kumari et al. (2023), Concept Confidence Score (CCS), and Concept Retrieval Score (CRS) Sharma et al. (2024). Additionally, we provide qualitative results for both the erased and retained concepts, allowing for a fair comparison of the various unlearning methods.

**Experiment Setting.** We assess the performance of concept erasure across the following categories: art style, logo, identity, and NSFW content, utilizing Stable Diffusion 1.4 (SD 1.4). The experiments were carried out with NVIDIA A6000 48GB GPU. We observe that different methods give optimal results at varying iterations. Therefore, we present the results at multiple iteration intervals to ensure a fair comparison.

**Quantitative Analysis.** We show the mean CLIP scores for the *8 erased concepts* from the SD 1.4 model in Table 1. The CLIP score is a metric used to measure how well the generated images semantically match their target concepts. For the *erased concepts*, lower CLIP score indicates a reduced similarity between the generated images and the given prompt, suggesting more effective concept erasure. For example, after erasing the concept "Amitabh Bachchan", the model would not generate images resembling to *Amitabh Bachchan*. In Table 1, JS method (ours) consistently outperforms CW (ours) and L2 (existing method) at steps 1500, 1400, and 1300. At step 1200, L2 is better, while at step 1100, CW gives better score. JS method exhibits particularly strong performance, achieving the lowest CLIP scores in the majority of the steps. Overall, we observe CW and JS either maintains similar performance or improves upon the L2 method.

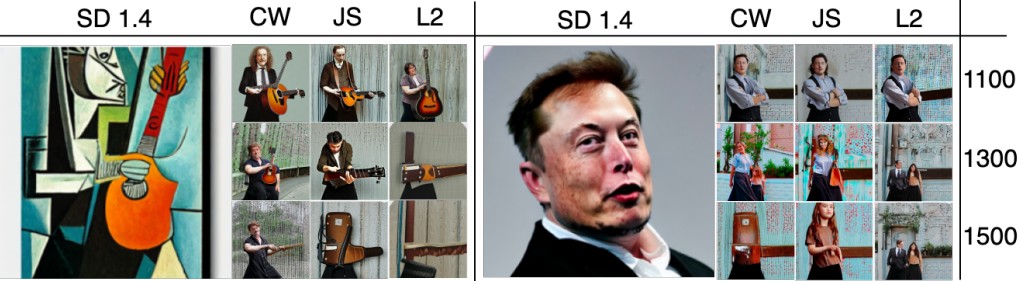

Figure 2: Comparing the unlearning results of CW, JS, and L2 after erasing the concepts of *Pablo Picasso* and *Elon Musk* from the SD 1.4. Prompts: *the old guitarist by Pablo Picasso, Elon Musk*. We observe that at 1100 step, both CW and JS generate old guitarist but without the style of Pablo Picasso but L2 generates a visibly young guitarist.

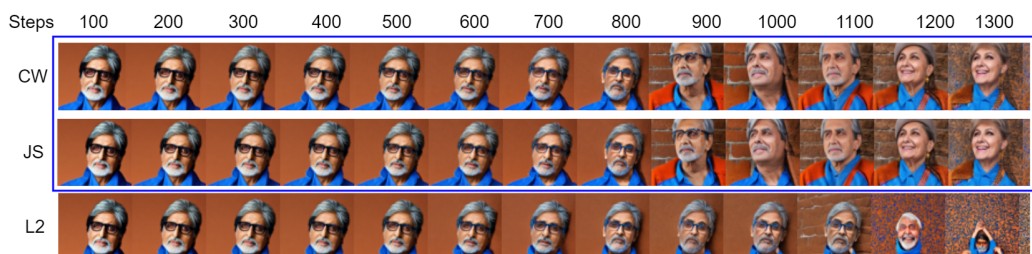

Figure 3: We show the progression of concept unlearning across iterations (100 to 1500) for CW, JS, and L2 methods.

We examine the CLIP directional similarity scores across various domains, with the results presented in Table 3. Five different concepts were selected, and we applied unlearning to the SD 1.4 model for each. The CLIP directional similarity scores for JS, CW, and L2 were then evaluated, where a higher score indicates more effective unlearning. For the concept of "baby," JS achieved the highest score of 0.1874. Similarly, CW obtained a score of 0.2085 when erasing the concept of "Elon Musk," outperforming both L2 and JS. On average, JS outperforms both CW and L2. Overall, the proposed JS and CW methods perform better than L2 in terms of the CLIP directional similarity score.

We evaluate the methods CW, JS and L2 based on the adversarial recovery attacks Concept Confidence Score (CCS), Concept Retrieval Score (CRS). We conduct the erasure for three distinct categories: *celebrity (Amitabh Bachchan)*, *object (baby)*, and *artistic style (vincent van gogh)*. We observe similar performance for all three methods CW, JS, and L2 in Table 2. This indicates that the change in loss function doesn't influence the ability of the unlearned model against adversarial recovery attacks such as CCS and CRS.

**Qualitative Analysis.** We show the visual results for erasing varieties of concepts: *artistic style*, *celebrity*, *baby/children*, *nudity*, *logo*, and *harmful contents*. In Figure 1, we show the unlearning results of CW, JS, and L2. In most of the concept erasure requests, we observe similar results except for erasure of *Nike* logo. In Nike concept erasure, CW and JS generate images of shoe with a different type of logo, whereas, L2 generates a footwear that is not a shoe but looks like crocs. We show another comparison between these methods in Figure 2. We erase the concepts of *Pablo Picasso* and *Elon Musk* from SD 1.4. We observe CW and JS generate old guitarist without the style of Pablo Picasso but L2 generates a young guitarist.

**Unlearning at Different Iterations.** Figure 3 illustrates the unlearning process across different iterations in JS, CW, and L2. The visuals provides insights into the dynamics of concept erasure when using different loss functions like JS, CW, and L2. After 800 steps, CW(ours) and JS(ours) generate different looking image than *Amitabh Bachchan* while L2 still generates an image similar to *Amitabh Bachchan*. In the final step (step 1500), L2 fails to generate human face while CW and JS generate human face that is not similar to *Amitabh Bachchan*.

**Visualization at the Teacher and Student Model.** Figure 4 compares the outputs of the teacher and student models during the unlearning process in JS, CW, and L2. The comparison highlights the

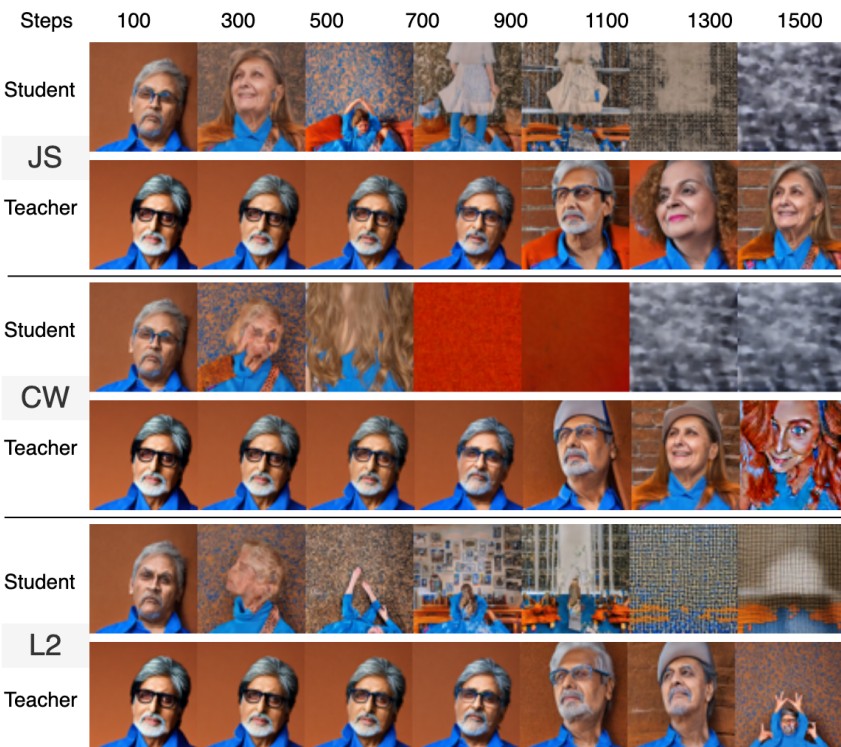

Figure 4: We compare the change in generated outputs by the teacher and student models in CW, JS, and L2. We show the outputs of teacher and student models at different iterations (100 to 1500) to assess the impact of using different loss functions.

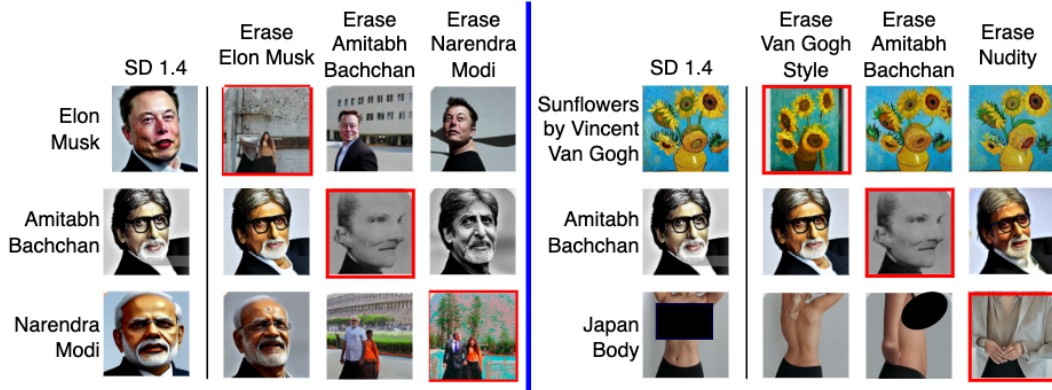

Figure 5: We show the results produced by the proposed method (JS) over the unlearned and retention set of concepts. The diagonal images are the generated for the *erasure concepts* and the remaining images are generated for the *retained concepts* (for the corresponding unlearned model). It shows our method preserves the general capabilities of the diffusion model after unlearning.

effectiveness of teacher-student framework in transferring the unlearning process while maintaining overall image quality. The student that is given an empty prompt is naturally producing noisy image for all the three methods. The teacher model which is eventually used for unlearning shows that JS and CW learn finer variations for the erasing concepts as compared to the L2 method.

**Observing the Results for the Retained Concepts.** To assess whether the unlearning methods maintain the model's overall capabilities post-unlearning, we present results on the retained concepts for the modified models. Figure 5 displays a diverse set of generated images unrelated to the erased concepts. The images along the diagonal represent the concepts meant to be removed, while the

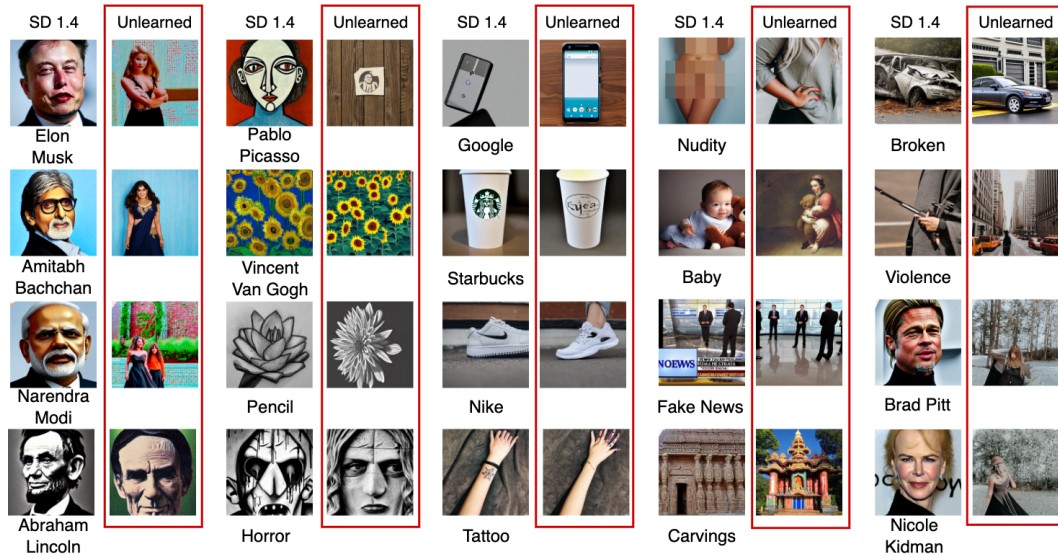

Figure 6: Before and after unlearning results for a variety of unlearning requests that cover the usecases for NSFW, restricted content, rights, and usage concerns. The results are shown for JS method.

remaining images correspond to the concepts the model should retain. It is evident that the proposed JS method effectively preserves the retained concepts while successfully forgetting the targeted ones.

**Concepts Generated Before and After Unlearning.** Figure 6 illustrates the results of our unlearning process across various concepts, both before and after applying the method. The comparison highlights the model's capability to effectively remove specific concepts while preserving overall image quality and coherence. The examples include facial unlearning (e.g., Elon Musk, Amitabh Bachchan, Narendra Modi, Brad Pitt, Nicole Kidman), artistic styles (e.g., Pablo Picasso, Vincent van Gogh, pencil art, tattoos), logos (e.g., Google, Starbucks, Nike), and other miscellaneous categories (e.g., nudity, children, fake news, temple carvings, violence, broken car). Additional results are shown in Figure 7, Figure 8, and Figure 9.

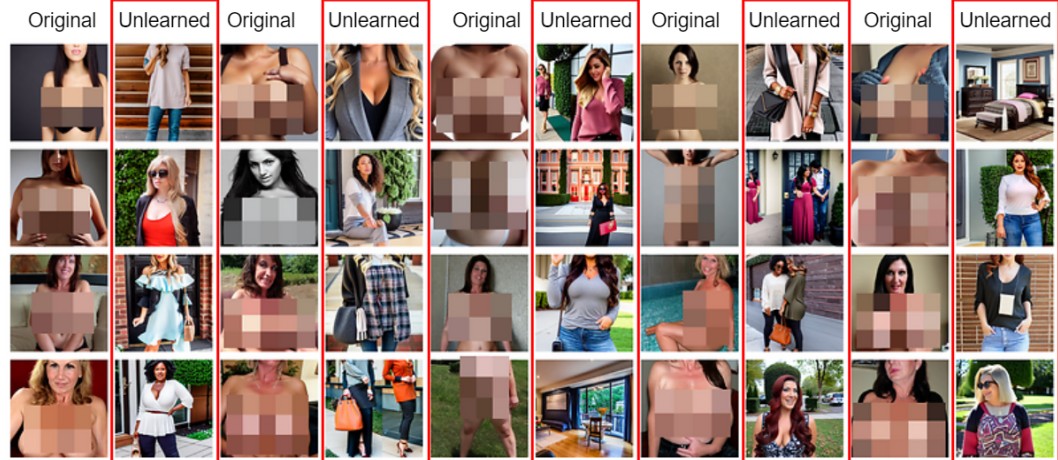

Figure 7: Before and after unlearning results while erasing the concept of *Nudity* (JS method).

## 6 CONCLUSION

In this paper, we proposed a set of robust methodologies for concept unlearning in diffusion models, utilizing advanced loss functions like Cramer-Wold distance and Jensen-Shannon (JS) divergence

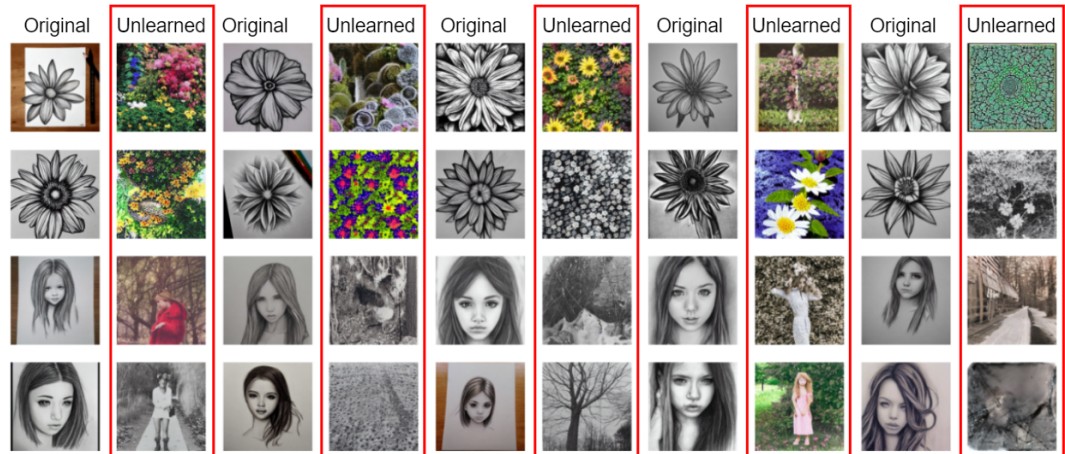

Figure 8: Before and after unlearning results while erasing the concept of *Pencil Art* (JS method).

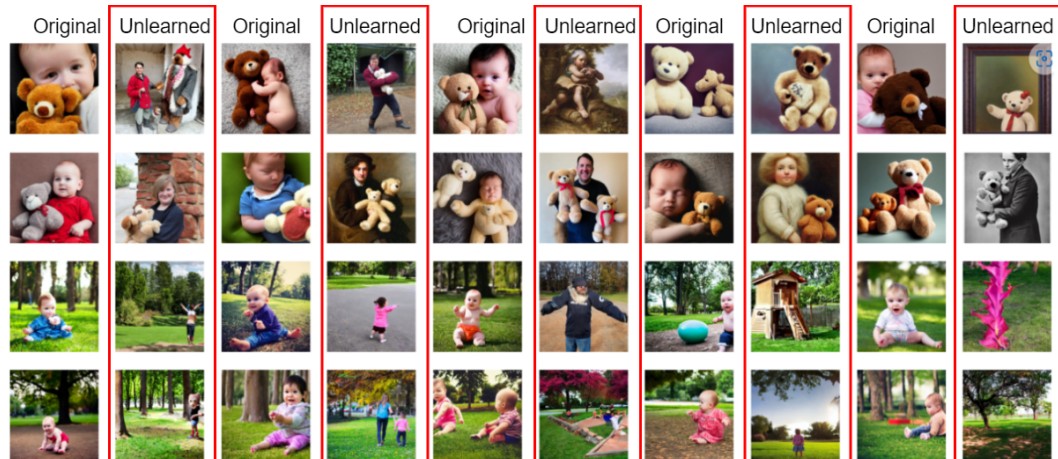

Figure 9: Before and after unlearning results while erasing the concept of *Baby* (JS method).

within a knowledge distillation framework. Our approach demonstrated improved performance over L2 loss based method over diverse concept removal categories. Extensive experiments and mathematical analysis confirmed the versatility and effectiveness of our unlearning method, paving the way for more controlled and responsible applications of text-to-image diffusion models.

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

# A APPENDIX

## A.1 PROOF OF LEMMA1

*Proof.* The L2 distance between $\mathbf{z}_T = [z_{T,1}, z_{T,2}, \ldots, z_{T,n}]$ and $\mathbf{z}_S = [z_{S,1}, z_{S,2}, \ldots, z_{S,n}]$ is given by:

$$\text{L2}(\mathbf{z}_T, \mathbf{z}_S) = \sqrt{\sum_{i=1}^{n} (z_{T,i} - z_{S,i})^2} \tag{4}$$

We can partition the sum into concept-related and non-concept-related dimensions:

$$\text{L2}(\mathbf{z}_T, \mathbf{z}_S) = \sqrt{\sum_{i \in \mathcal{D}_c} (z_{T,i} - z_{S,i})^2 + \sum_{i \in \mathcal{D}_{nc}} (z_{T,i} - z_{S,i})^2} \tag{5}$$

$$\tag{6}$$

If the concept $\mathbf{c}$ is unlearned, then $z_{T,i} = z_{S,i}$ for all $i \in \mathcal{D}_c$. Hence:

$$\sum_{i \in \mathcal{D}_c} (z_{T,i} - z_{S,i})^2 = 0. \tag{7}$$

However, if $z_{T,i} \neq z_{S,i}$ for any $i \in \mathcal{D}_{nc}$, then:

$$\sum_{i \in \mathcal{D}_{nc}} (z_{T,i} - z_{S,i})^2 > 0, \tag{8}$$

implying:

$$\text{L2}(\mathbf{z}_T, \mathbf{z}_S) > 0 \tag{9}$$

Thus, L2 distance is not zero even if the concept is fully unlearned, as long as there are differences in irrelevant dimensions. $\square$

## A.2 PROOF OF LEMMA2

*Proof.* Let the concept $\mathbf{c}$ be represented as a linear combination of dimensions $i, j, k$ such that:

$$\mathbf{c} = \alpha z_{T,i} + \beta z_{T,j} + \gamma z_{T,k}. \tag{10}$$

Suppose unlearning results in a proportional decrease in these dimensions in $\mathbf{z}_S$, i.e.,

$$z_{S,i} = z_{T,i} - \delta, \quad z_{S,j} = z_{T,j} - \delta, \quad z_{S,k} = z_{T,k} - \delta, \tag{11}$$

for some constant $\delta$. The L2 distance is:

$$\text{L2}(\mathbf{z}_T, \mathbf{z}_S) = \sqrt{(z_{T,i} - (z_{T,i} - \delta))^2 + (z_{T,j} - (z_{T,j} - \delta))^2 + (z_{T,k} - (z_{T,k} - \delta))^2} = \delta\sqrt{3} \tag{12}$$

However, the true change in the concept is along $\mathbf{c}$, and the L2 distance does not reflect the conceptual change accurately unless $\alpha = \beta = \gamma$. Therefore, L2 distance fails to correctly capture the change in correlated dimensions. $\square$

## A.3 PROOF OF LEMMA3

*Proof.* Let's recall the definition of JS Divergence 1. *Definition of JS Divergence*: Let $P = [p_1, p_2, \ldots, p_n]$ and $Q = [q_1, q_2, \ldots, q_n]$ be the probability distributions derived from the teacher and student representations such that:

$$\sum_{i=1}^{n} p_i = 1, \quad \sum_{i=1}^{n} q_i = 1, \quad p_i, q_i \geq 0 \tag{13}$$

The Jensen-Shannon (JS) divergence between $P$ and $Q$ is defined as:

$$\text{JS}(P, Q) = \frac{1}{2} KL(P\|M) + \frac{1}{2} KL(Q\|M), \tag{14}$$

where $M = \frac{P+Q}{2}$ and $KL$ is the Kullback-Leibler divergence:

$$KL(P\|M) = \sum_{i=1}^{n} p_i \log \frac{p_i}{m_i}, \quad KL(Q\|M) = \sum_{i=1}^{n} q_i \log \frac{q_i}{m_i}, \quad m_i = \frac{p_i + q_i}{2} \tag{15}$$

2. *Partition into Concept and Non-Concept Dimensions*: Let $\mathcal{D}_c$ denote the set of concept-related dimensions and $\mathcal{D}_{nc}$ denote the irrelevant dimensions. Decompose $P$ and $Q$ as:

$$P = [P_c, P_{nc}], \quad Q = [Q_c, Q_{nc}], \tag{16}$$

where $P_c = \{p_i : i \in \mathcal{D}_c\}$ and $P_{nc} = \{p_i : i \in \mathcal{D}_{nc}\}$ (similarly for $Q_c$ and $Q_{nc}$). If $P_{nc} = Q_{nc}$, then $p_i = q_i$ for all $i \in \mathcal{D}_{nc}$. For these dimensions, the KL divergence terms are zero:

$$\sum_{i \in \mathcal{D}_{nc}} p_i \log \frac{p_i}{m_i} = 0, \quad \sum_{i \in \mathcal{D}_{nc}} q_i \log \frac{q_i}{m_i} = 0 \tag{17}$$

Thus, the JS divergence simplifies to:

$$\text{JS}(P, Q) = \frac{1}{2} \sum_{i \in \mathcal{D}_c} p_i \log \frac{p_i}{m_i} + \frac{1}{2} \sum_{i \in \mathcal{D}_c} q_i \log \frac{q_i}{m_i} \tag{18}$$

3. *Invariant to Irrelevant Changes*: For $i \in \mathcal{D}_{nc}$, if $p_i = q_i$, the divergence between $P$ and $Q$ in these dimensions will remain zero, regardless of the magnitude of $p_i$ and $q_i$. In contrast, L2 distance will still consider the differences in $\mathcal{D}_{nc}$, leading to misleading conclusions about whether the concept has been unlearned.

4. *Effectiveness in Concept Unlearning*: If unlearning is successful and $P_c = Q_c$, then:

$$\text{JS}(P, Q) = 0, \tag{19}$$

even if $P_{nc} \neq Q_{nc}$. Thus, JS divergence accurately measures concept alignment by focusing only on relevant dimensions, whereas L2 distance remains sensitive to changes in irrelevant dimensions.

□

### A.4 PROOF OF LEMMA4

*Proof.* 1. *Definition of Cramér-Wold Distance*: The Cramér-Wold distance between two distributions $P$ and $Q$ is defined as the supremum over all 1-dimensional linear projections:

$$\text{CW}(P, Q) = \sup_{\|\theta\|=1} \|P_\theta - Q_\theta\|, \tag{20}$$

where $\theta$ is a unit vector, and $P_\theta$ and $Q_\theta$ are the 1-dimensional projections of $P$ and $Q$ along $\theta$:

$$P_\theta = \theta^T \mathbf{z}_T, \quad Q_\theta = \theta^T \mathbf{z}_S. \tag{21}$$

2. *Effectiveness in Capturing Correlations*: If the concept $\mathbf{c}$ is represented by a set of correlated dimensions $\mathcal{D}_c$, define a linear combination for the concept:

$$\mathbf{c} = \sum_{i \in \mathcal{D}_c} \alpha_i z_{T,i}. \tag{22}$$

Consider a projection $\theta$ such that it aligns with the concept subspace. If $P$ and $Q$ are identical along $\theta$, i.e., $P_\theta = Q_\theta$, then:

$$\|P_\theta - Q_\theta\| = 0. \tag{23}$$

3. *Detecting Higher-Order Correlations*: Unlike L2 distance, which measures dimension-wise differences, Cramér-Wold distance takes into account joint distributions and higher-order correlations. Thus, if unlearning leads to a change in joint correlations but not in individual dimensions, L2 distance might be zero, while CW distance will detect the conceptual change.

4. *Invariant to Irrelevant Projections*: If $\theta$ is orthogonal to the concept subspace, then $P_\theta = Q_\theta$ for all such projections, even if $\mathbf{z}_T \neq \mathbf{z}_S$ in irrelevant dimensions. Thus, Cramér-Wold distance provides a more comprehensive measure of concept unlearning by considering projections along all directions, ensuring that the concept is completely removed. □

### A.5 MORE VISUAL RESULTS

We show additional visual results of concept erasure in the proposed JS method in Figure 14(erasing *Nike* logo), Figure 15(erasing *Narendra Modi*), Figure 16(erasing *Amitabh Bachchan*), Figure 17(erasing *Elon Musk*), Figure 18(erasing *child*), Figure 19(erasing *Vincent Van Gogh styled paintings*), Figure 20(erasing *Pablo Picasso styled paintings*).

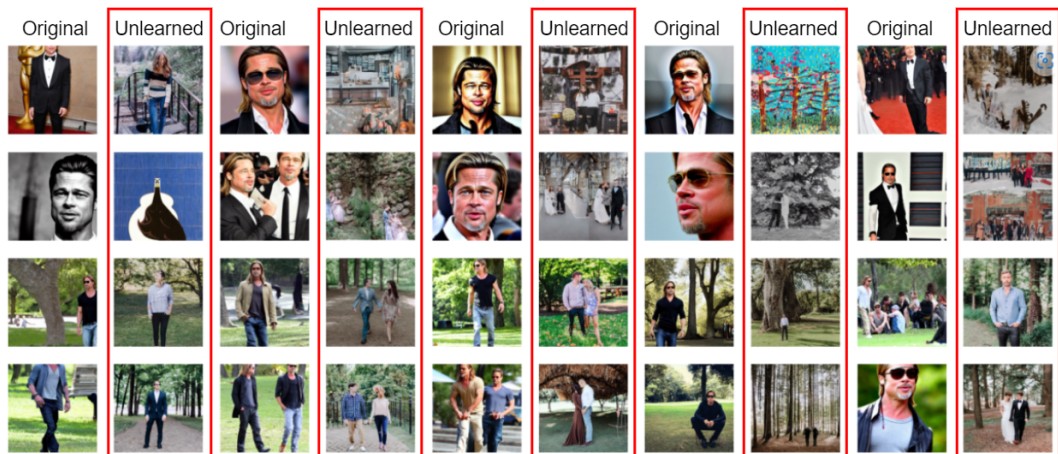

Figure 10: Unlearning results after erasing the concept of *Brad Pitt* (JS method).

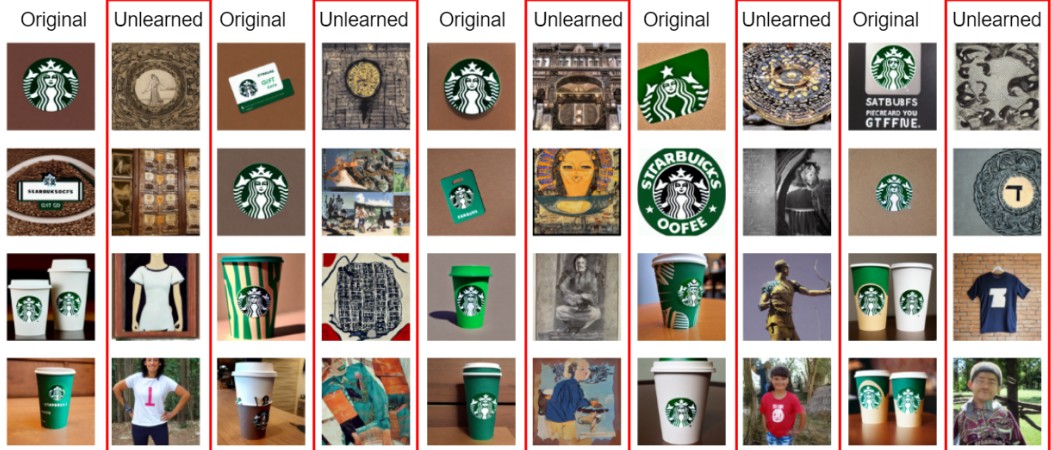

Figure 11: Unlearning results after erasing the concept of *Starbucks* logo (JS method).

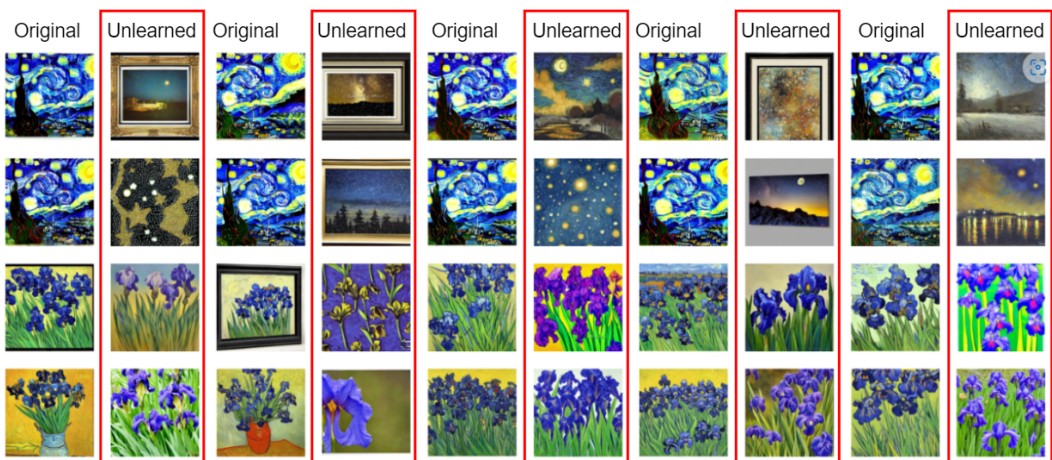

Figure 12: Unlearning results after erasing the concept of *Vincent Van Gogh* (JS method).

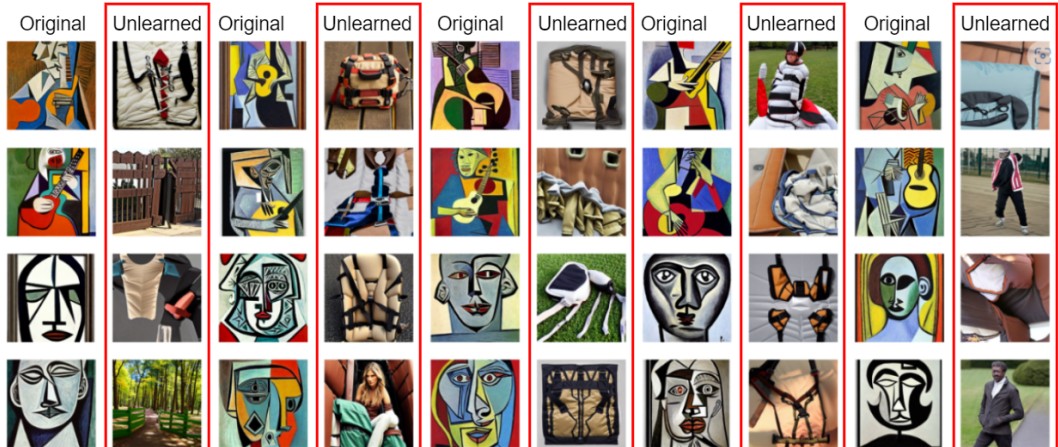

Figure 13: Unlearning results after erasing the concept of *Pablo Picasso* (JS method).

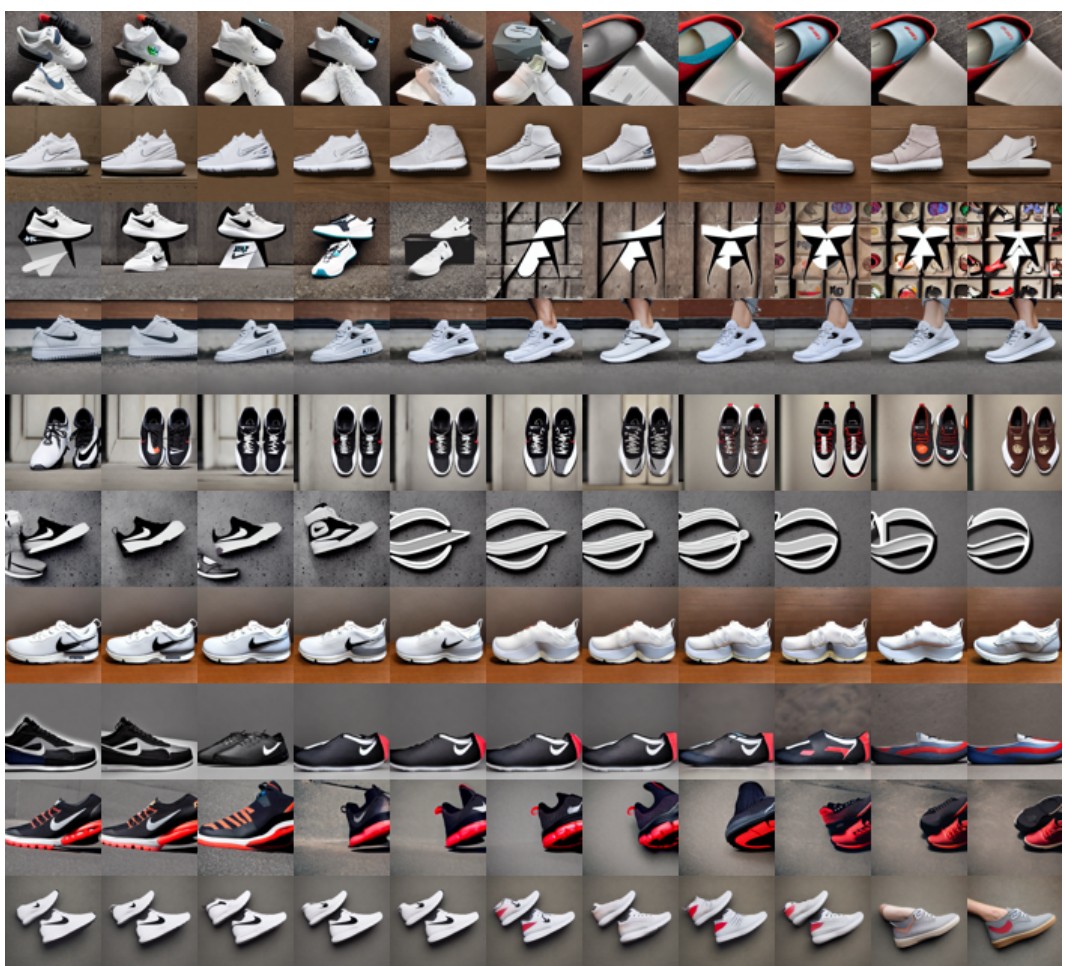

Figure 14: Unlearning results after erasing the concept of *Nike* logo (JS method).

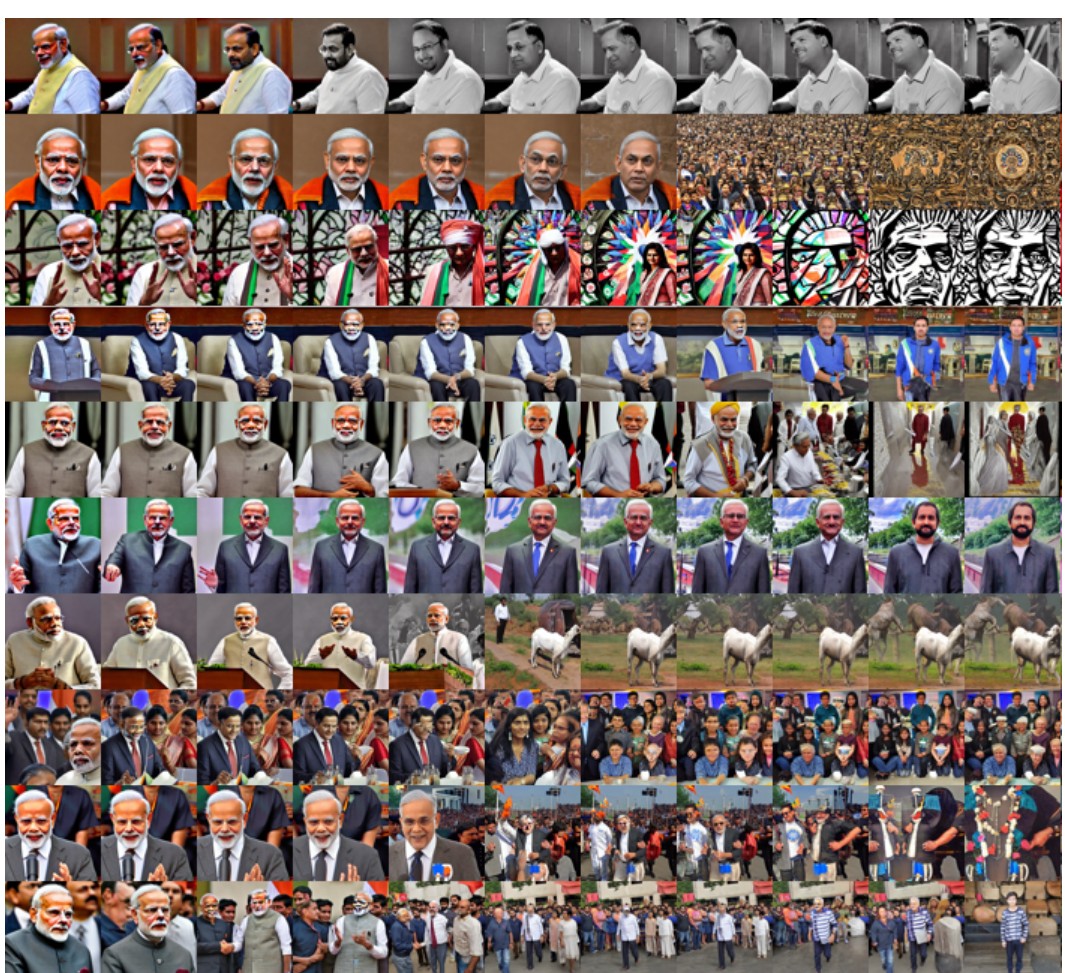

Figure 15: Unlearning results after erasing the concept of *Narendra Modi* (JS method).

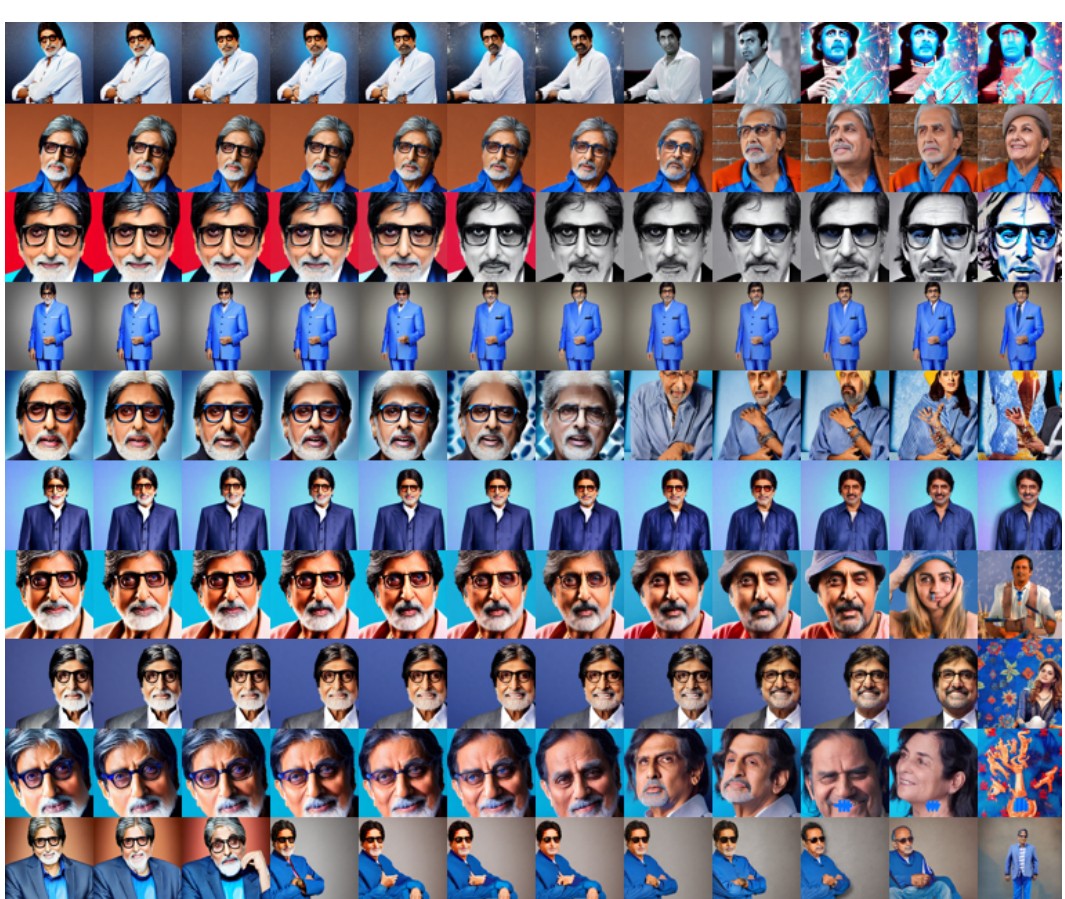

Figure 16: Unlearning results after erasing the concept of *Amitabh Bachchan* (JS method).

972
973
974
975
976
977
978
979
980
981
982
983
984
985
986
987
988
989
990
991
992
993
994
995
996
997
998
999
1000
1001
1002
1003
1004
1005
1006
1007
1008
1009
1010
1011
1012
1013
1014
1015
1016
1017
1018
1019
1020
1021
1022
1023
1024
1025

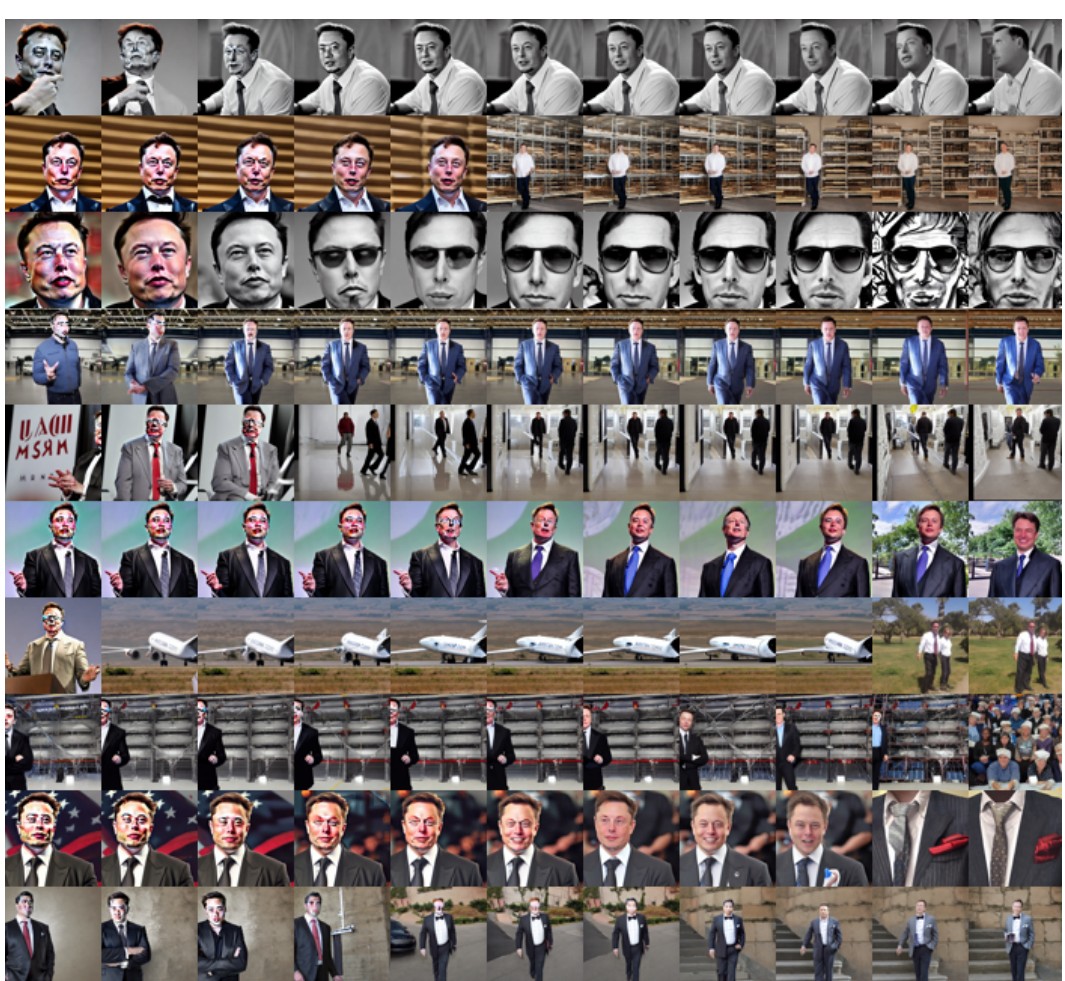

Figure 17: Unlearning results after erasing the concept of *Elon Must* (JS method).

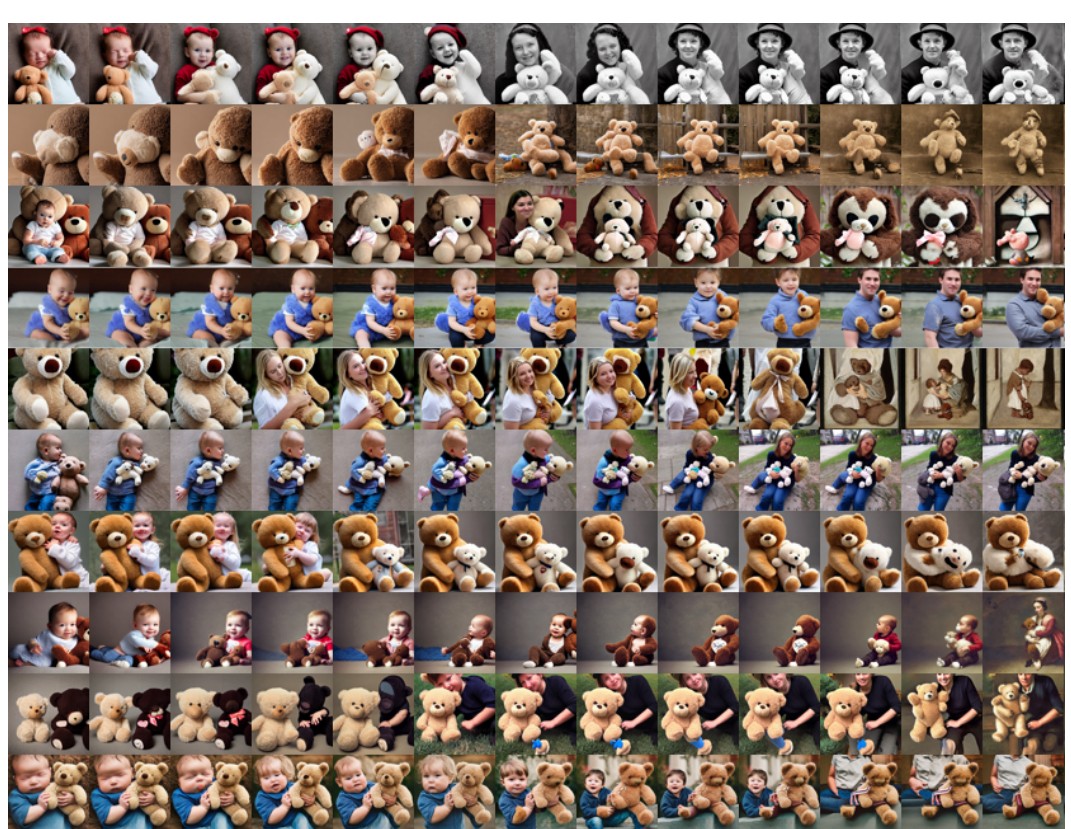

Figure 18: Unlearning results after erasing the concept of *Child* (JS method).

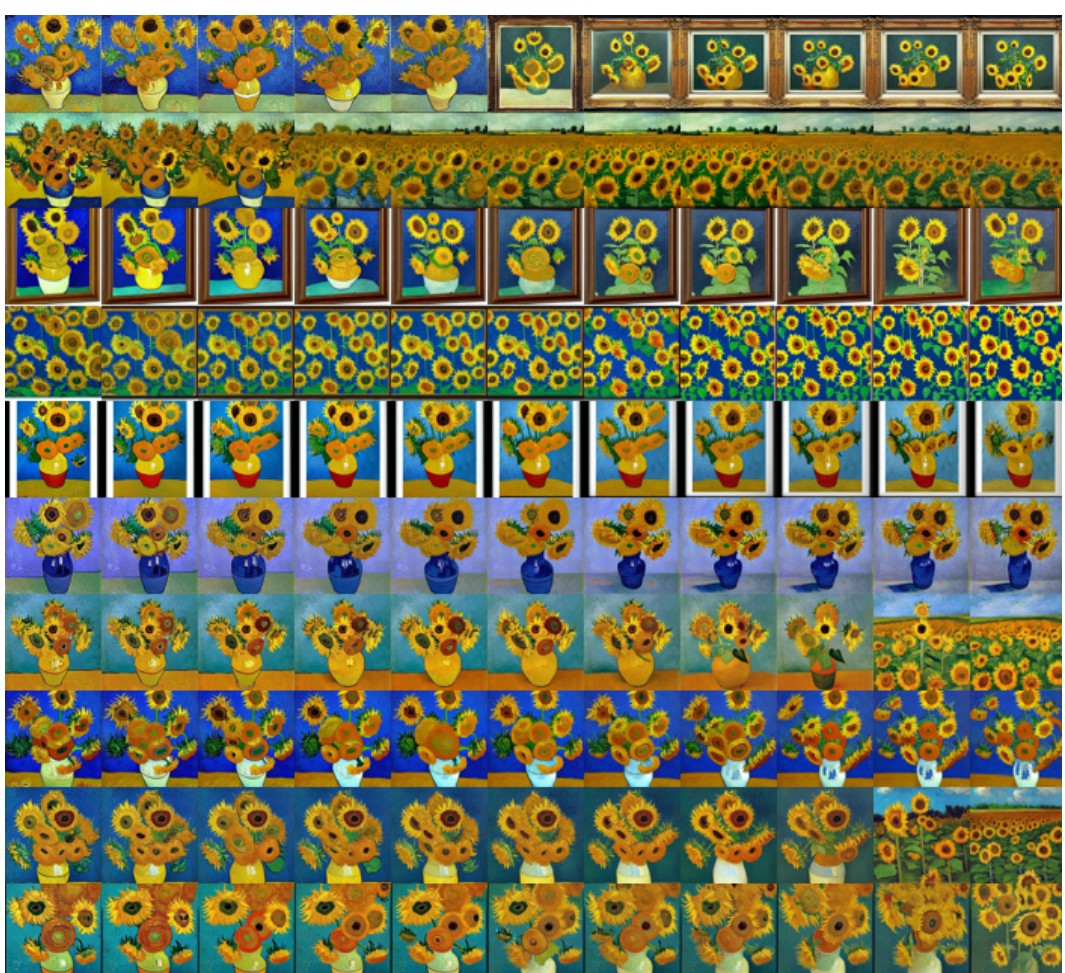

Figure 19: Unlearning results after erasing the concept of *Vincent Van Gogh* (JS method).

Figure 20: Unlearning results after erasing the concept of *Pablo Picasso* (JS method).

