# OpenReview forum: "Towards Robust Concept Erasure in Diffusion Models: Unlearning Identity, Nudity and Artistic Styles"
_ICLR.cc/2025/Conference — ICLR 2025 Conference Withdrawn Submission_

### Official Review · Reviewer_KETW · 2024-10-28

**Soundness:** 2
**Presentation:** 1
**Contribution:** 2
**Rating:** 3
**Confidence:** 4

**Summary:**

The paper presents Jensen Shannon divergence and Cramer-Wold Distance as alternatives for L2 loss in diffusion model unlearning paradigm. They claim that both JS and CW facilitate in versatile and efficient unlearning. Through well defined theoretical proofs and experimental evaluations, they show their approach is effective in erasing compared to traditional L2 losses.

**Strengths:**

The paper is well motivated. Especially the arguments around JS and Cramer-Wold distance.

**Weaknesses:**

1. The related works section can take the opportunity to potentially provide a more "state-of-the-art" view of erasing techniques like:
a. UCE https://openaccess.thecvf.com/content/WACV2024/papers/Gandikota_Unified_Concept_Editing_in_Diffusion_Models_WACV_2024_paper.pdf
b. MACE https://openaccess.thecvf.com/content/CVPR2024/papers/Lu_MACE_Mass_Concept_Erasure_in_Diffusion_Models_CVPR_2024_paper.pdf


2. SDD is used as the base erasure technique in the paper, but is not explained in the related works or background. The authors could explain the method briefly for clarity.

3. The method is well motivated - but the evaluations are not sufficient quantitatively.
a. table 1 presents that the JS is better at certain epochs of training and not so good in others. This is a confusing evaluation for comparison. Ideally - these methods can be trained for much longer and compared at the final epochs. I am curious why authors chose this
b. Table 2 - the unlearning performance seems to be very similar across all the methods (which does not support the claim). Also the test set seems to be a single prompt which isn't an ideal setup.
c. Table 3 is only tested on a single prompt which is not ideal sample set
d. if the student model is not being trained - why is the output changing for both teacher and student in Figure 4?


4. The motivation of the paper comes from 4 lemmas - these were not empirically tested in the paper

**Questions:**

1. The main precedence of L2 objective being "unreliable" is that even when concepts dimensions are erased, L2 is non-zero because of other dimensions. Is this under assumption that there are certain dimensions that represent concept and certain that do not? Is this a practical assumption? I could be misunderstanding the lemma 1 proof. I understand that the diffusion loss is being computed in the noise space (where certain pixels may belong to the concepts and certain pixels do not) - but isn't this being taken care by the ground truth in ESD or CA or any other prior work?

2. In Section 3.2 - does Eq. 3 mean to optimize the student or the teacher? This is used interchangeable across the paper - The authors could clarify this.

3. If the student model is not being trained - why is the output changing for both teacher and student in Figure 4?

4. Why did the authors compare the methods at certain unlearning epoch steps? Why not completely unlearn and test against the final version?

5. How does the performance look across a larger dataset?

---

### Official Review · Reviewer_bTAh · 2024-10-31

**Soundness:** 3
**Presentation:** 3
**Contribution:** 2
**Rating:** 5
**Confidence:** 4

**Summary:**

This paper addresses the issue of catastrophic forgetting in diffusion models when unlearning specific concepts, a problem that arises with the traditional L2 loss approach. To mitigate this risk, the paper proposes using JS divergence and Cramer-Wold distance as alternative strategies. It claims that JS divergence and CW distance allow for a more robust alignment of the distribution in the diffusion model’s latent space, thereby minimizing interference with non-targeted concepts. Experimental results demonstrate that this approach not only outperforms L2-based methods in concept removal but also reduces the risk of catastrophic forgetting.

**Strengths:**

- The paper raises issues with existing unlearning techniques and proposes a convincing method to address them effectively.

-  The paper effectively demonstrates the validity of using CW distance and JS divergence as knowledge distillation losses by mathematically proving that these approaches mitigate catastrophic forgetting.

**Weaknesses:**

- Among the experiments, there is no quantitative evidence showing that knowledge distillation with CW distance and JS divergence results in less catastrophic forgetting compared to L2 loss. Additionally, Figure 5 only shows generation results unrelated to the erased concepts when using JS; there are no results for L2 loss, making comparison difficult.

- Since this approach was tested only on SDD among various knowledge distillation-based unlearning methods, it remains unclear whether it can be generally applied to other methods.

- In Table 3, CW distance and JS divergence do not consistently outperform L2, making it difficult to conclude that CW and JS are superior to L2.

**Questions:**

- It is unclear from the paper whether CW and JS outperform L2 in terms of catastrophic forgetting. A quantitative metric to measure the extent of catastrophic forgetting would allow for a helpful comparison between CW, JS, and L2 losses.

- I would like to see quantitative results on the use of JS and CW, instead of L2 loss, for knowledge distillation-based unlearning methods such as ESD-u, ESD-x [A], and Ablating concepts [B]. If this approach could be applied more generally across other methods, it would make the paper’s claims more convincing.

- I would like to know about the details of the benchmarks used in the experiments. Additionally, I believe that more diverse concepts should be tested in the quantitative evaluation. In Table 3, performance is reported only for 'baby,' 'Narendra Modi,' 'Elon Musk,' 'Amitabh Bachchan,' and 'Nike,' raising concerns about possible cherry-picking.

[A] Erasing concepts from diffusion models. In Proceedings of the IEEE/CVF International Conference on Computer Vision, pp. 2426–2436, 2023.
[B] Ablating concepts in text-to-image diffusion models. In Proceedings of the IEEE/CVF International Conference on Computer Vision, pp. 22691–22702, 2023.

---

### Official Review · Reviewer_CBas · 2024-11-03

**Soundness:** 2
**Presentation:** 3
**Contribution:** 3
**Rating:** 5
**Confidence:** 4

**Summary:**

The paper proposes a method to selectively erase sensitive concepts from text-to-image diffusion models to improve ethical use and content moderation. By using advanced loss functions—Jensen-Shannon divergence and Cramér-Wold distance—instead of traditional L2 loss, the method achieves more effective and nuanced unlearning. Through empirical tests and theoretical analysis, the approach demonstrates improved performance.

**Strengths:**

1) The paper employs a teacher-student knowledge distillation framework to transfer unlearning from a student model back to the teacher model.
2) The authors provide extensive experimental results across multiple concepts and include visual comparisons that clearly illustrate the method’s impact.
3) The paper tackles a real-world challenge that holds ethical and legal importance, especially given the potential misuse of generative models for harmful content.

**Weaknesses:**

1) The authors argue that the commonly used L2 loss function is suboptimal for unlearning, leading them to propose the use of Jensen-Shannon (JS) divergence and Cramér-Wold (CW) distance. However, the justification for using these specific loss functions could be more rigorous. While the paper briefly explains why these alternatives might work better, it lacks an in-depth theoretical or empirical foundation demonstrating why these loss functions are inherently superior for the range of concept erasure scenarios tested.

2) Although the paper claims that the proposed method outperforms existing techniques, the comparative analysis appears to be limited. The choice of baseline methods seems selective, focusing on techniques that primarily use L2 loss, while more diverse and recent alternatives in the literature may have been overlooked. Additionally, the experiments are conducted on the Stable Diffusion 1.4 model, and it is unclear how these results generalize to other diffusion models or versions. Testing on multiple models or providing a discussion on potential limitations across models would enhance the validity of the claimed generalizability.

3) The use of a teacher-student knowledge distillation framework is central to the method, but there is limited discussion about the assumptions and limitations of this setup. For example, the model may experience "catastrophic forgetting," where learning new concepts could unintentionally erase other important knowledge. Although the authors mention mitigation of catastrophic forgetting, the specifics of how they address it are vague, and empirical evidence on this aspect seems sparse.

**Questions:**

1) Why were JS divergence and Cramér-Wold distance chosen over other distance measures? Could other loss functions like Wasserstein distance or mutual information provide similar or better results?
2) What are the limitations of using a teacher-student distillation framework for concept unlearning? How does the distillation process handle "catastrophic forgetting" or unintended interference with non-targeted knowledge?
3) The paper mentions preserving retained concepts post-unlearning. However, does the model’s creative diversity or ability to generalize decrease when multiple concepts are erased consecutively?

---

### Note · Authors · 2024-11-30

**Comment:**

We feel that substantial amount of experiments are required to support the claims in the paper. The same has been suggested by the Reviewers. Therefore, we withdraw the paper. We thank all the Reviewers to give their valuable time to review this paper.

**Withdrawal Confirmation:**

I have read and agree with the venue's withdrawal policy on behalf of myself and my co-authors.